# A Transcription Regulatory Sequence in the 5′ Untranslated Region of SARS-CoV-2 Is Vital for Virus Replication with an Altered Evolutionary Pattern against Human Inhibitory MicroRNAs

**DOI:** 10.3390/cells10020319

**Published:** 2021-02-04

**Authors:** Manijeh Mohammadi-Dehcheshmeh, Sadrollah Molaei Moghbeli, Samira Rahimirad, Ibrahim O. Alanazi, Zafer Saad Al Shehri, Esmaeil Ebrahimie

**Affiliations:** 1La Trobe Genomics Research Platform, School of Life Sciences, College of Science, Health and Engineering, La Trobe University, Melbourne, VIC 3086, Australia; m.mohammadidehcheshmeh@latrobe.edu.au or; 2School of Animal and Veterinary Sciences, The University of Adelaide, Adelaide, SA 5371, Australia; 3Department of Animal Science, College of Agricultural and Life Sciences, University of Wisconsin, Madison, WI 1675, USA; molaeimoghbe@wisc.edu; 4Department of Medical Genetics, National Institute of Genetic Engineering and Biotechnology, Tehran 1497716316, Iran; samirad1731@gmail.com; 5National Center for Biotechnology, Life Science and Environment Research Institute, King Abdulaziz City for Science and Technology (KACST), Riyadh 6086, Saudi Arabia; ialenazi@kacst.edu.sa; 6Department of Medical Laboratories, College of Applied Medical Sciences, Shaqra University, KSA, Al dawadmi 1678, Saudi Arabia; zaf@su.edu.sa; 7School of BioSciences, The University of Melbourne, Melbourne, VIC 3052, Australia

**Keywords:** COVID-19, biomarker, drug repurposing, microRNA, SARS-CoV-2 replication, microRNA vaccine, nanoparticle vaccine, variant discovery

## Abstract

Our knowledge of the evolution and the role of untranslated region (UTR) in SARS-CoV-2 pathogenicity is very limited. Leader sequence, originated from UTR, is found at the 5′ ends of all encoded SARS-CoV-2 transcripts, highlighting its importance. Here, evolution of leader sequence was compared between human pathogenic and non-pathogenic coronaviruses. Then, profiling of microRNAs that can inactivate the key UTR regions of coronaviruses was carried out. A distinguished pattern of evolution in leader sequence of SARS-CoV-2 was found. Mining all available microRNA families against leader sequences of coronaviruses resulted in discovery of 39 microRNAs with a stable thermodynamic binding energy. Notably, SARS-CoV-2 had a lower binding stability against microRNAs. hsa-MIR-5004-3p was the only human microRNA able to target the leader sequence of SARS and to a lesser extent, also SARS-CoV-2. However, its binding stability decreased remarkably in SARS-COV-2. We found some plant microRNAs with low and stable binding energy against SARS-COV-2. Meta-analysis documented a significant (*p* < 0.01) decline in the expression of MIR-5004-3p after SARS-COV-2 infection in trachea, lung biopsy, and bronchial organoids as well as lung-derived Calu-3 and A549 cells. The paucity of the innate human inhibitory microRNAs to bind to leader sequence of SARS-CoV-2 can contribute to its high replication in infected human cells.

## 1. Background

Coronavirus genomes are single-stranded mRNAs, containing both coding and untranslated regions (UTRs). The 5′UTR and 3′UTR are crucial for coronavirus RNA replication, transcription, and dominating host systems biology [1]. However, their exact roles and their evolutions, specifically in the new SARS-CoV-2, are mainly unknown. 

There are some 5′UTR regions that are vital for virus replication (Figure 1). A specific region of 70 nucleotides at the 5′ end of the genome, referred as the ‘leader’ sequence, has been found at the 5′ ends of all encoded transcripts, highlighting its importance. A conserved cis-acting element, named transcription regulatory sequence (TRS), immediately follows the leader sequence, representing a unique feature of coronaviruses. 

Compared to the other positive-strand RNA viruses, coronaviruses have a unique and complicated pattern of continuous and discontinuous RNA synthesis (Figure 2). Like the other positive-strand RNA viruses, continuous RNA synthesis happens for genome replication to yield multiple copies of the genomic RNA. In continuous RNA synthesis, a full-length complementary negative-strand RNA is utilized as the template for generation of progeny virus genomes [2]. In discontinuous transcription, during the synthesis of sub-genomic negative-strand RNAs, a premature termination and template switch occur to add copies of the leader sequence [2]. The presence of the 5′ leader sequence provides an efficient strategy for the efficient accumulation of coronaviruses mRNAs and proteins during infection because of the protection of viral mRNAs from endonucleolytic cleavage of the capped mRNAs [3]. Furthermore, the complement of the leader sequence supports initiation of positive-strand RNA synthesis, generating negative-strand sub-genomic RNAs as templates for further productions of positive-strand sub-genomic mRNAs [2].

Whereas UTR is a non-coding region, another unique structure of coronaviruses is the existence of a short open reading frame (ORF) on one of the 5′UTR stem loops. In bovine coronavirus, maintenance of the short ORF is positively correlated with viral RNA accumulation [4]. The other non-coding region, 3′-UTR folds into a unique stem-loop two motif secondary structure that is required for virus viability.

UTRs are also important in the context of host microRNA interaction. MicroRNAs play their negative regulatory roles via sequence-specific interactions with the 3′ or 5′UTRs [5]. Consequently, sequence variation in the viral UTR regions can prevent the binding of human microRNAs, so preventing microRNA-based immunity [6,7]. The problem is more serious in case of SARS-CoV-2 which originated recently from bat, such that human microRNAs have not evolved to interact with their UTRs. 

Here, we hypothesize that the TRS and leader sequences of 5′ UTRs are crucial for coronavirus RNA replication. Moreover, microRNAs can bind to the TRS and leader sequences of the 5′UTR to reduce SARS-CoV-2 replication. The first aim of this study was to unravel the evolution of the TRS and leader sequences in 5′UTR regions of SARS-CoV-2. The second aim was to identify inhibitory microRNAs, present in human and other organisms, that can bind to the key regulatory regions of SARS-CoV-2 sequences. 

## 2. Methods

### 2.1. Data Collection

Complete genomes of SARS-COV-2, SARS, MERS, bat coronavirus, and bovine coronavirus, including complete sequences of non-coding regions, were downloaded from The National Center for Biotechnology Information (NCBI, www.ncbi.nlm.nih.gov). The sequences were: Severe acute respiratory syndrome coronavirus 2 isolate Wuhan-Hu-1 (NC_045512), SARS coronavirus Tor2 (NC_004718), Tylonycteris bat coronavirus HKU4 isolate CZ01 (MH002337), Middle East respiratory syndrome-related coronavirus isolate NL13892 (MG987420), and bovine coronavirus isolate BCoV-ENT (NC_003045).

### 2.2. Inhibitory MicroRNA Prediction 

Available microRNA information of human and other organisms, deposited in TargetScan [8], were downloaded. In total, the information of 9994 microRNAs, including sequences of seed region and mature sequences, were obtained. The seed region is a conserved sequence with the average length of 6–8 bp [9]. The seed region is critical for binding and hybridization of the microRNA to its target. Appendix A represents the seed region, mature sequence, microRNA family, and the associated MiRBase IDs of the 9994 microRNAs. 

Then, the reverse complement sequence of UTR regions of SARS-COV-2, SARS, MERS, and bovine coronavirus were searched against the known seed sequences to find the microRNAs with affinity binding to the specific regions in 5′UTR, TRS, and leader sequences by TargetScan. 

Based on the MiRBase IDs, obtained from the TargetScan, fasta format of mature sequences of microRNAs were retrieved from the miRBase database [10]. Using the RNAhybrid v2.2 database, the binding probability of mature microRNAs against TRS and leader sequences of coronaviruses were evaluated through calculating of a minimum free energy hybridization [11]. The thermodynamic binding energy of −15 kcal/mol was used as the minimum cut-off, and binding energy of −25 kcal/mol pointed the stable binding [12].

### 2.3. Sequence Alignment between 5′UTR of Coronavirus and MicroRNA Seed Regions and Phylogenetic Trees

Multiple sequence alignment was performed using CLUSTALW algorithm by CLC Genomics Workbench 20 (QIGEN, Venlo, The Netherlands). Phylogenetic trees, based on the maximum likelihood phylogeny approach, were constructed using UPGMA method, nucleotide substitution model of Jukes Cantor, gamma distribution parameter of 1.0, and bootstrap of 1000.

### 2.4. Literature-Mining Based Drug Repurposing 

Pathway Studio Database (Elsevier) was employed to find the drugs, food supplements, and chemicals (small molecules) with promoting effects on 5′ UTR inhibitory microRNAs, as described recently [13]. To perform literature mining, MedScan tool was used that employs the NLP algorithm to mine extract relations from biomedical texts, mainly PubMed [14]. In addition to the mined sentences of relations, Medscan also records the title of literature, authors, the year of publication, Medline (PubMed) reference number, and type of relation. The results are deposited in Mammalian + ChemEffect + DiseaseFx database of Pathway Studio which is enriched by a range of extra systems biology information like the subcellular location and protein class (such as receptor, ligand, transcription factor, small RNA, small molecule, etc.), from Gene Ontology Consortium, as well as KEGG pathways. The database updates using cloud technology by addition of new mined relationships and entities from recent publications. Statistics of Mammalian + ChemEffect + DiseaseFx database used for literature mining is presented at Table 1.

In short, a highly enriched database with more than one million chemicals, 138,000 proteins, and 12 million protein interactions were employed for drug repurposing. 

### 2.5. Multivariate Analysis

Thermodynamic binding energy values (kcal/mol) between microRNAs and 5′UTR regions of coronaviruses were used as input for PCA and clustering analysis. Analysis was performed in MINITAB 18 (https://www.minitab.com). Graphs were visualized by GraphPad Prism 7 (GraphPad Software, Inc. California, CA, USA). Correlation matrix was used for PCA analysis. Clustering was performed based on Euclidean distance matrix and average linkage method.

### 2.6. MIR-5004 Expression Analysis in Response to SARS-CoV-2 Infection: Meta-analysis Approach

In a comprehensive study, the expression of MIR-5004 was evaluated in COVID-infected samples compared to non-infected ones in human and ferret. The raw RNA-seq reads in FASTQ format were downloaded from NCBI SRA (https://www.ncbi.nlm.nih.gov/sra) using CLC Genomics Workbench v.20.0 (QIAGEN, Venlo, The Netherlands). In total, 42 infected and non-infected samples of trachea, lung biopsy, bronchial organoids, lung-derived Calu-3 cells, lung alveolar A549 cells, and lung epithelium NHBE cells, from GSE150819, GSE147507, and GSE159522 were utilized (Table 2).

FastQC tool (http://www.bioinformatics.babraham.ac.uk/projects/fastqc/) was used to assess quality raw. MIR-5004 genomic sequence, NR_049800, was downloaded from NCBI. Mapping of short reads to MIR-5004 reference genome was performed using CLC Genomics Workbench based on the following mapping parameters: Mismatch cost = 2, Insertion cost = 3, Deletion cost = 3, Length fraction = 0.1, Similarity fraction = 0.6, and Maximum number of hits for a read = 10. Mapped reads to MIR-5004 reference genome (mapped reads per million) were used as expression measurement.

Comprehensive Meta-analysis (CMA) software Version 3.3.070 (Biostat Inc., Englewood, NJ, USA) [15] was used for meta-analysis of MIR-5004 expression in response to SARS-CoV-2 infection in different studies and different tissue types (trachea, lung biopsy, bronchial organoids, Calu-3 cells, and A549 cells, and NHBE cells). Fixed standardized effects model was employed for meta-analysis [16].

### 2.7. Variant Discovery on Genomic Sequence of Hsa-MIR-5004-3p, 5′UTR Inhibitory MicroRNAs, as COVID-19 Risk Factors

The human genetic variation genetic variation database of NCBI (dbSNP database) [17] was employed as the main resource for variants gathering. The Pathway Studio tool was used for retrieving hsa-MIR-5004-3p genomic variants by mining more than two hundred million deposited variants in dbSNP and 1000 Genomes project. Genomic locations of variants were recorded (CDs, 3′UTR, 5′UTR, intergenic, or intronic variants). Then, translational impact of the identified hsa-MIR-5004-3p variants including missense, splice disrupt, CDs indel, nonsense, misstart, and non-stop were determined. Finally, the proportion of each type of variant was calculated.

To evaluate the possible functional impact of the identified hsa-MIR-5004-3p variants, the Genomic Evolutionary Rate Profiling (GERP)++ conservation score was used. GERP is an evolutionary conservation score which have a good correspondence with clinical significance and pathogenicity level [18]. GERP++ demonstrates the constrained elements in multiple alignments by quantifying substitution deficits. These deficits identify substitutions that would have happened if the element were neutral DNA but did not happen as the element has been experienced functional constraint. Low values of GERP++ score stand for low level of conservation and high values for high level of conservation.

## 3. Results

### 3.1. Comparative Analysis of the 5′UTR of Human Pathogenic and Non-Pathogenic Coronaviruses

Our primary analysis of reference sequences distinguished a pattern of evolution in the leader sequence and TRS of SARS-CoV-2, by comparison with MERS and the bovine coronavirus (Figure 3). Interestingly, the TRS sequence is identical in all coronaviruses that infect human (SARS-CoV-2, SARS, and MERS) which differs in non-human pathogens, such as bovine coronavirus (Figure 3). In other words, TRS can explain the host range of a coronavirus.

### 3.2. Identifying the MicroRNAs that Can Bind to the Leader Sequence and TRS of SARS-CoV-2 (5′UTR Inhibitory MicroRNAs)

Mining microRNAs against the leader sequence of SARS-CoV-2 resulted in the discovery of 39 microRNAs with an acceptable thermodynamic binding energy (cut-off of less than minus 15 kcal/mol). Table 3 presents a list of microRNAs and their thermodynamic binding energy that potentially bind to at least one type of coronavirus (SARS-COV-2, SARS, MERS, bat coronavirus, or bovine coronavirus). ptc-MiR474b, ptc- MiR474a, csa-let-7d, cin-let-7d-5p, csi-miR3953, and gga-MiR-6608-3p were microRNAs with stable thermodynamic binding energy (lower than −22 kcal/mol) against leader sequence of SARS-CoV-2. 

hsa-MIR-5004-3p was the only human microRNA able to target the leader sequence of SARS and SARS-CoV-2. However, its binding stability decreased remarkably in SARS-COV-2 (−19.4 kcal/mol), compared with SARS-COV-2 (−25.9 kcal/mol) (Table 3 and Figure 4). Notably, our analysis showed that the leader sequence of SARS-COV-2 is mutated (insertion type mutation, CA), thus escaping microRNA-RNA hybridization (Figure 5). Lack of innate human inhibitory microRNAs able to bind to SARS-COV-2 contributes to the high replication of SARS-CoV-2 in infected human cells. 

### 3.3. The Leader Sequence of SARS-CoV-2 Has a Unique Pattern of MicroRNA Binding, Compared with SARS, MERS, Bat, and Bovine Coronaviruses

Multivariate analysis of thermodynamic binding energy values of mined inhibitory microRNAs (Table 3) demonstrates a unique pattern of SARS-CoV-2 evolution (Figure 6). 

PCA discriminated efficiently between SARS-CoV-2 and other types of coronaviruses where PCA1 and PCA2 described 71.9% of variation in the data (Appendix A). SARS-CoV-2 had negative values of both PCA1 and PCA2 (Figure 6). Interestingly, hsa-miR-5004-3p is one of the top five important microRNAs in PCA1 with an absolute coefficient > 0.2 (Appendix A). ptc-miR474b, cme-miR1863, bta-miR-2284ab, and bdi-miR5065 were the top microRNAs in PCA2. Consistent with this finding, in comparison with the other human pathogens (SARS and MERS), the leader sequence of SARS-COV-2 has a remarkably higher binding energy against microRNAs (−16.33 kcal/mol against 18.58 kcal/mol and −18.34 kcal/mol, respectively) (Table 3). Higher binding energy results in lower stability of microRNA-5′UTR binding and provides this opportunity for SARS-COV-2 to escape the inhibitory microRNAs. 

Clustering shows that SARS-CoV-2 has only 41.6% similarity with SARS in its pattern of binding to microRNAs (Appendix A and Figure 6). 

### 3.4. Drug Repurposing to Induce 5′UTR Inhibitory MicroRNAs

As presented in Figure 7, we developed a literature mining-based drug repurposing approach to identify inhibitory microRNAs against the leader sequence of SARS-CoV-2. To this end, more than 1 million drugs and small molecules, and 12 million relations (binding, biomarker, expression, chemical reaction, promoter binding, microRNA effect, etc.) were mined by natural language processing (NLP).

Literature mining-based drug repurposing could not identify any drug, small molecule, or food supplement with direct interaction with hsa-miR-5004-3p (Figure 7). However, downregulation of KLF4, by heme and calcium, sowed indirect potential to upregulate hsa-miR-5004-3p as an inhibitory microRNA against SARS-CoV-2. Heme is an iron-containing tetradentate ligand. 

### 3.5. Significant Decline in Expression of MIR-5004 after SARS-COV-2 Infection

Meta-analysis of MIR-5004 expression in a range of tissues and cells (42 samples), including trachea, lung biopsy, lung-derived Calu-3 cells, and lung alveolar A549 cells, highlighted a significant (*p* < 0.01) decline in expression in of MIR-5004 after SARS-COV-2 infection (Figure 8). Interestingly, the decline in expression of MIR-5004-3p was more significant in trachea and lung tissues than in cell lines. Noticeably, the decline in expression of human inhibitory microRNAs with increasing age, and in a range of diseases such as diabetes and obesity, has been observed [19] that can contribute to higher mortality of SARS-CoV-2 in elderly patients.

### 3.6. hsa-miR-5004-3p Genomic Variation 

Impaired 5′UTR inhibitory microRNAs in the human genome could account for the high rate of virus replication in human cells. Consequently, mutations in genomic sequences in the 5′UTR inhibitory microRNAs may be considered as a risk factor of COVID-19 infection. Mining more than two hundred million deposited variants in dbSNP, using the Pathway Studio tool (Elsevier), resulted in discovery of 9 variants in hsa-MIR-5004 (Table 4). Six of these variants were splice-disrupted mutations with possible regulatory functions. Table 4 presents the potentially damaging variants according to their GERP++ conservation scores. The GERP++ conservation scores vary from −12.3 to 6.17. Larger values demonstrate higher conservation. Splice-disrupted mutations in MIR5004 have high GERP++ scores.

## 4. Discussion

UTRs, particularly in the 5′UTR, are of high translational importance. The leader sequence and TRS in the 5′ non-coding part of SARS-CoV-2 can be considered as the Achilles’ heel of SARS-CoV-2. Leader sequence is located at the 5′ ends of all encoded transcripts, highlighting its potential significance. The TRS can explain the host range and pathogenicity of a coronavirus. UTRs are potential sites for antiviral drugs to bind and inhibit the virus replication. There is no report on disruption of 5′UTR in SARS-CoV-2 but, in bovine coronavirus, it has been found that disruption of either stem-loop III or stem-loop IV of the 5′UTR stops virus RNA replication, suggesting that these regions function as cis-acting elements [4,20]. On the other hand, microRNAs that bind to SARS-CoV-2 UTRs can be induced by drugs or food supplements to reduce virus replication. Enhancing host microRNA defense machinery against the 5′UTR region of a virus can help in prevention of SARS-CoV-2 infection. The above-mentioned strategies represent potentially achievable treatments against COVID-19 infection. In this study, we presented a model of literature mining-based drug discovery to induce inhibitory microRNAs against leader sequence of SARS-CoV-2.

Some host cell microRNAs are important components of the host immune defense against viral infection as they destroy the viral RNA [21]. In this study, we found that hsa-miR-5004-3p is a unique human microRNA with the ability to target the leader sequence of SARS and SARS-CoV-2. Our comprehensive meta-analysis also documented a significant (*p* < 0.01) decline in the expression of MIR-5004-3p after SARS-COV-2 infection in trachea, lung, and bronchial organoids as well as in lung-derived Calu-3 and A549 cells. Interestingly, a decreased level of hsa-miR-5004-3p has been reported in the whole blood of patients during dengue virus (DENV) infection. hsa-miR-5004-3p was undetectable in early DENV infection, but expression was high in most of the healthy controls and recovered dengue patients. This finding demonstrated suppression of hsa-miR-5004-3p during the early phases of DENV infection and its importance in patient recovery [22]. Like COVID-19, DENV is a rampant arboviral illness worldwide [22] with high viral transmission success and immune evasion. 

We found a significant trend in the 5′UTR of SARS-CoV-2 to escape from binding of hsa-miR-5004-3p by insertion-type mutation. Such mutations decrease microRNA-5′UTR binding stability and allow SARS-CoV-2 to escape the available human microRNA immunity system. We suggest that the lack of innate human inhibitory microRNAs for SARS-CoV-2 contributes to its high replication in the infected human cells. On the other hand, mining of two hundred million deposited human genomic variants led us to discover splice-disrupted mutations in the genomic structure of hsa-miR-5004-3p. These mutations can negatively affect hsa-miR-5004-3p function in preventing SARS-CoV-2 replication. 

The abundance of some human inhibitory microRNAs against SARS-CoV-2 is associated with age [19]. It has been discussed that higher COVID-19 virulence in the aged patients [23] is potentially related to the decline in microRNAs in elderly people [19]. In addition to age, some of the human microRNAs downregulate in coronary artery disease, kidney disease, colorectal cancer, osteosarcoma, prostate cancer, obesity/diabetes, myocardial injury, hepatocellular carcinoma, non-small cell lung cancer, gastrointestinal tumors, and colorectal cancer [19]. Like age, the mentioned conditions can also contribute to lower abundance of human inhibitory microRNAs and consequently, higher severity and mortality of COVID-19. This hypothesis needs to be examined in future studies.

The 5′UTR inhibitory microRNAs, such as hsa-miR-5004-3p or those of plant origin such as ptc-miR474 and csi-miR3953, have promising potential for developing new therapies. MicroRNA vaccine, nanoparticles, synthesized microRNAs, microRNA exosome, and dietary microRNAs can be considered in this context. MicroRNA-peptide fusion is also suggested as potential vaccine candidates [24]. In addition to hsa-miR-5004-3p, plant microRNAs with very low binding energy (less than −22 kcal/mol) against leader sequence of 5′UTR of SARS-CoV-2, such as ptc-miR474b and ptc-miR474a (*Populus trichocarpa*) and csi-miR3953 (*Citrus sinensis*) are possible candidates for inclusion. Nanoformulations of the SARS-CoV-2 inhibitory microRNAs can successfully and safely deliver microRNAs to lung cells, to reduce viral replication in the host cell and suppress the viral load [25]. MicroRNA-based nanoparticles can also be utilized in the form of nanovaccines to prevent SARS-CoV-2 infection. Compared with traditional vaccines, nanovaccines have many potential benefits, including: (1) Specific targeting to infection sites, (2) minimal off-target effects, and (3) the possibility of nasal spray/drop formulation for direct activation of the immune system in the respiratory tract [25]. Due to shortness of microRNA sequences (18–25 nucleotides), synthesizing of inhibitory microRNAs are readily conducted. It has been suggested that mixtures of synthesized microRNAs, called multiple microRNA cocktail, targeting different coding (such as S gene) and UTR regions of SARS-CoV-2 can be considered as new COVID-19 treatment strategy [19,26]. This synthesized mixture may be delivered to the target host cells by liposomes like exosome (extracellular vesicle), polymer-based carriers, or nanoparticles [26]. High potential of microRNA exosomes as biomarkers of infection and recovery in COVID 19 has been suggested [24]. On the other hand, the combination of a mRNA vaccine with the immune promoting agents, such as inhibitory microRNAs, can increase its efficiency and protection, even in the case of new SARS-CoV-2 mutations.

Interestingly, expression of microRNAs, complementary to the 3′UTR of viruses, is the main protecting strategy in plants that degrades the viral RNA or blocks its translation by ribosomes [26]. Plants are a rich source for microRNAs, with remarkable therapeutic and preventive roles in many diseases [27]. Recently, it has been suggested that microRNAs in food can be absorbed by the human circulatory system [28]. The absorbed microRNA, named dietary microRNA, can regulate gene expression and biological processes in mammalian cells [28]. Dietary microRNA, as a novel functional feature of food [28], opens a new potential avenue for safe and accessible COVID19 protection and treatment. Further research on plant dietary microRNAs may lead to a safe treatment strategy against COVID19 by oral use of plant microRNAs or whole plants containing SARS-CoV-2 inhibitory microRNAs.

Sequence variation in the non-coding viral genomic regions may predispose people to develop more severe disease. It should be noted that the genetic signature of the pathogenesis severity in the non-coding regions of SARS-CoV-2 is still unknown. A recent report on human–mink–human transmission cycle [29] highlights the necessity to uncover all functional mutations, including the ones that occur in UTR regions. The availability of 212,346 (at 21 November 2020) SARS-CoV-2 genomic sequences, including 159,057 sequences of the full genome and high coverage, in GISAID (https://www.gisaid.org/) [30] and NCBI provides the chance of pattern recognition in 5′UTR sequences of SARS-CoV-2, particularly against host microRNA inhibitory machinery, by machine learning models. Models and statistics such as decision tree classification based on association rule mining and deep learning [31,32,33,34,35,36] that have been used for eukaryotic promoter and UTR analysis, can be examined for UTR analysis of SARS-CoV-2. It should be noted that some of the sequences that have been deposited in GISAID as full genomes have incomplete or low-quality sequencing in 3′UTR and 5′UTR regions. Data cleaning of 5′UTR and 3′UTR regions is a major task before analysis as many of the announced SARS-CoV-2 full genomes with complete coding sequences have incomplete, short, or low-quality UTR sequences. This problem casts doubt on the reliability of the currently identified SARS-CoV-2 sequences and UTR mutations.

## 5. Conclusions

Non-coding regions are crucial for SARS-CoV-2 replication, transcription, and domination of host systems biology. Unravelling the reasons underpinning SARS-CoV-2 success in dominating human cells and its high transmission rate is crucial for future research. The present study is a significant step towards unravelling the evolution of the 5′UTR in SARS-CoV-2, discovering the key regions, and utilizing the UTRs for lowering virus load in the infected cells. We have shown that human hsa-miR-5004-3p and several plant microRNAs are good microRNA candidates to target the leader sequence of SARS-CoV-2 at 5′UTR region. 

In this study, an evolutionary pattern in the 5′UTR of SARS-CoV-2 from SARS was discovered whereby SARS-CoV-2 tries to escape hsa-miR-5004-3p binding by the generation of insertion-type mutations. The lack of a human inhibitory microRNAs to target the 5′ UTR of SARS-CoV-2 can contribute in immunity evasion and pathogenesis in SARS-CoV-2. Decline in expression of human inhibitory microRNAs with increasing age, and in a range of diseases such as diabetes and obesity, can also contribute to higher mortality of SARS-CoV-2 in elderly patients and in individuals suffering from these diseases. 

We also developed a literature mining-based drug repurposing strategy to induce inhibitory microRNAs that are potentially active against the leader sequence of SARS-CoV-2. Activation of inhibitory machinery microRNAs by drug repurposing and food supplements are easily achievable treatment strategies against COVID-19. Additionally, the microRNAs identified can be utilized for other therapeutic strategies such as development of microRNA vaccines, nanoparticles, synthesized microRNAs, microRNA exosome, and dietary microRNA.

## Figures and Tables

**Figure 1 cells-10-00319-f001:**
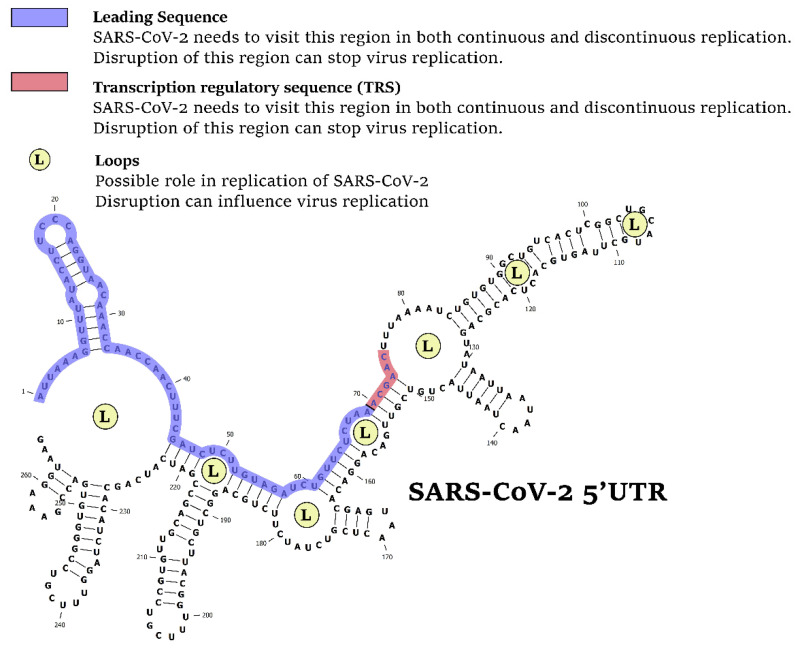
The 5′UTR is the Achilles’ heel of SARS-CoV-2. Disruption of specific regions in 5′UTR, including leading sequence, transcription regulatory sequence, or loops (L) can stop the virus replication.

**Figure 2 cells-10-00319-f002:**
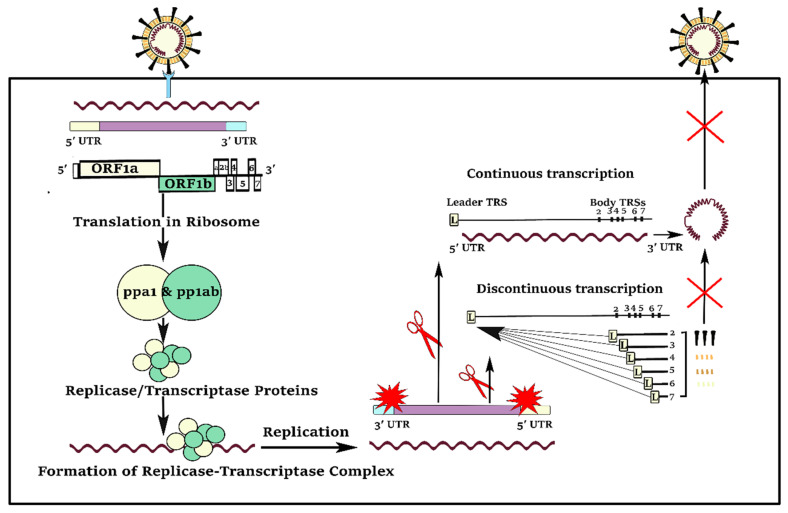
In both continuous and discontinuous RNA synthesis patterns, SARS-CoV-2 virus segments need to visit the 5′UTR, particularly their transcription regulatory sequence (TRS) and leader sequence (L). The 5′UTR is a good target to reduce virus load in the host cell.

**Figure 3 cells-10-00319-f003:**
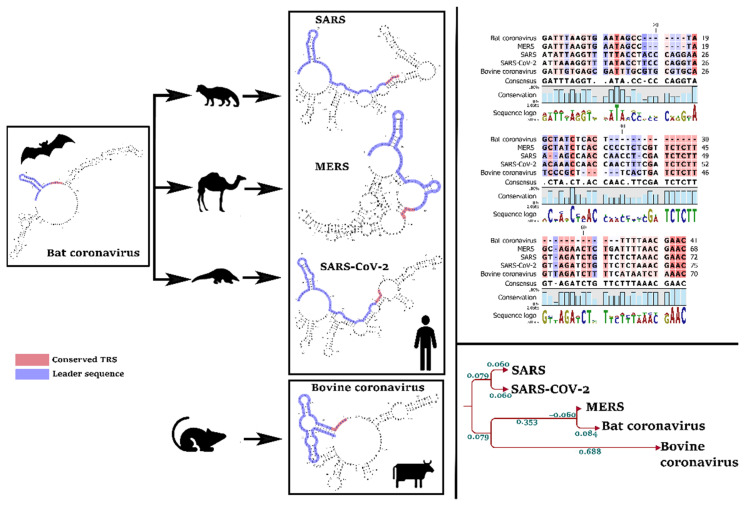
Evolution of “leader sequence” and TRS (transcription regulatory sequence) in the 5′UTR of human pathogens (SARS, MERS, and SARS-CoV-2) and non-human pathogens (bovine). We found a significant evolution pattern in SARS-CoV-2 compared with the other coronaviruses.

**Figure 4 cells-10-00319-f004:**
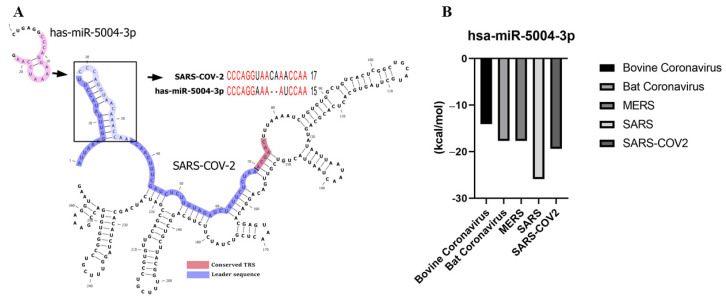
(**A**) hsa-miR-5004-3p is the only human microRNA able to target the leader sequence of SARS-CoV-2. (**B**) Binding energy (Kcal/mol) between leader sequence of coronaviruses and hsa-miR-5004-3p. Lower binding energies (smaller values) demonstrate higher binding stability between microRNA and leader sequence.

**Figure 5 cells-10-00319-f005:**
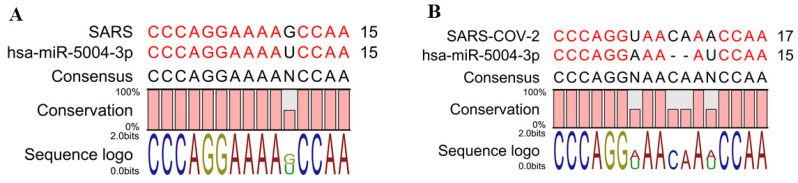
A two-nucleotide insertion mutation in the leader sequence of SARS-COV-2 results in lower binding between the viral leader sequence and seed region of hsa-miR-5004-3p. (**A**) Alignment of hsa-miR-5004-3p with SARS. (**B**) Alignment of hsa-miR-5004-3p with SARS-CoV-2.

**Figure 6 cells-10-00319-f006:**
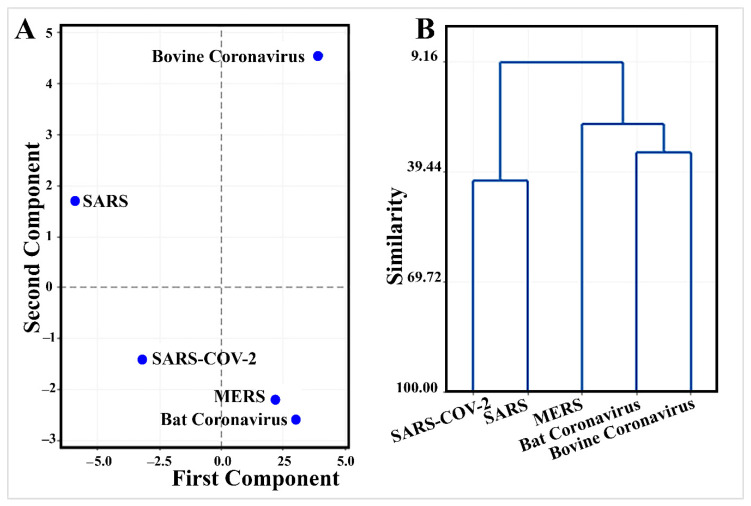
Multivariate analysis of thermodynamic binding energy values of the leader sequence of coronaviruses against mined inhibitory microRNAs (Table 3) demonstrates a unique pattern for SARS-CoV-2 evolution. (**A**) Principal component analysis discriminates SARS-CoV-2 from other coronaviruses. (**B**) Clustering shows that SARS-CoV-2 has only 41.6% similarity to SARS in its pattern of binding to microRNAs.

**Figure 7 cells-10-00319-f007:**
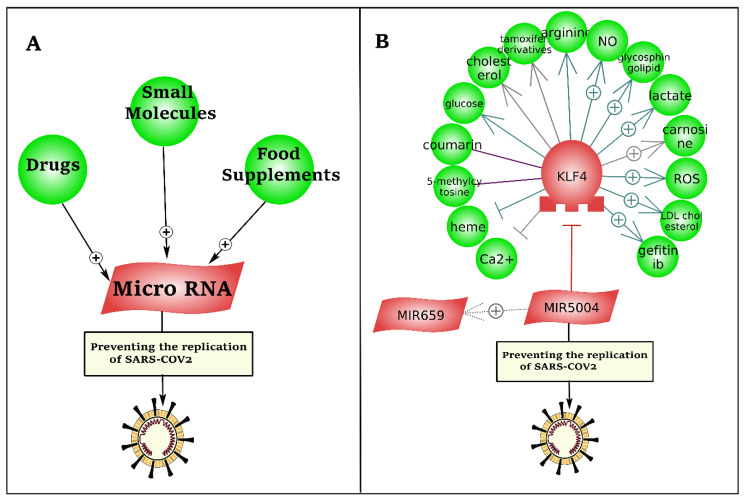
Drug repurposing to induce inhibitory microRNAs against the leader sequence and TRS of the 5′UTR region of SARS-CoV-2. (**A**) The presented pipeline of literature mining-based drug discovery. (**B**) Upregulation of hsa-miR-5004-3p has potential to prevent SARS-CoV-2 replication via binding to the leader sequence in the 5′UTR of virus.

**Figure 8 cells-10-00319-f008:**
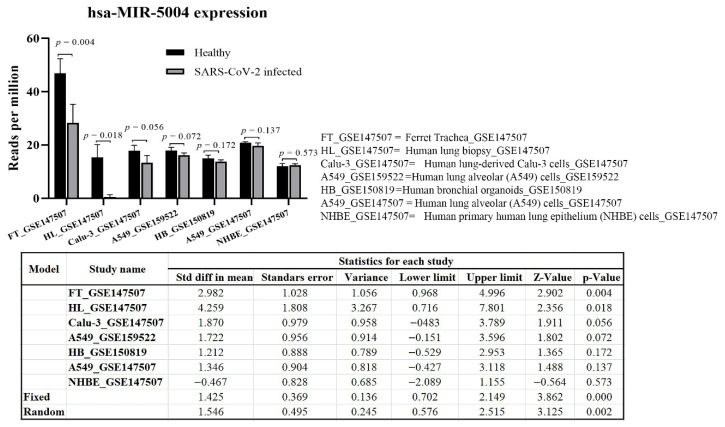
hsa-MIR-5004 expression response to SARS-CoV-2 infection. Meta-analysis revealed a significant (*p* < 0.01) decline in expression of hsa-MIR-5004 after SARS-COV-2 infection. hsa-MIR-5004 expression was studied in 42 samples of trachea, lung biopsy, lung-derived Calu-3 cells, and lung alveolar A549 cells before and after SARS-CoV-2 infection.

**Table 1 cells-10-00319-t001:** Statistics of Mammalian + ChemEffect + DiseaseFx database used for literature mining in this study (September 2020).

Entities	Number	Relations	Number
Small molecules (including drugs)	1,053,259	Binding	1,123,702
Protein	138,106	Biomarker	120,448
Cell process	9771	Cell expression	1,213,035
Cell Object	607	Chemical reaction	58,888
Cells	4155	Clinical trial	109,386
Clinical parameters	5126	Direct regulation	766,707
Complex	998	Expression	832,784
Diseases	20,855	Functional associations	1,775,463
Functional class	5489	Genetic change	379,261
Genetic Variant	127,872	Molsynthesis	160,178
Organ	3839	Moltransport	251,347
Treatments	78	Promoter binding	44,619
Tissue	574	Protein modification	73,859
Total number of entities	1,370,729	Quantitative change	421,884
		Regulation	5,193,796
		State change	128,112
		MicroRNA effects	57,743
		Total number of relations	12,653,469

**Table 2 cells-10-00319-t002:** Samples (n = 42) used for MIR-5004 expression analysis against SARS-CoV-2 infection.

Experiment ID	Sample ID (NCBI)	Organism	Tissue/Cell Line	SARS-CoV-2 Infected/Non-Infected	Total Number of Reads	SARS-CoV-2 Strain
GSE150819	SRR11811019	Human	Lung bronchial organoids	Non-infected	32,214,210	Non-infected (mock)
	SRR11811020	Human	Lung bronchial organoids	Non-infected	32,443,162	Non-infected (mock)
	SRR11811021	Human	Lung bronchial organoids	Non-infected	33,310,500	Non-infected (mock)
	SRR11811022	Human	Lung bronchial organoids	Infected	31,662,278	SARS-CoV-2/Hu/DP/Kng/19-020
	SRR11811023	Human	Lung bronchial organoids	Infected	35,953,491	SARS-CoV-2/Hu/DP/Kng/19-020
	SRR11811024	Human	Lung bronchial organoids	Infected	32,416,198	SARS-CoV-2/Hu/DP/Kng/19-020
GSE147507	SRR11517725-28	Human	human lung biopsies	Non-infected	57,660,692	Non-infected (mock)
	SRR11517729-32	Human	human lung biopsies	Non-infected	40,524,836	Non-infected (mock)
	SRR11517733-36	Human	human lung biopsies	Infected	10,561,476	USA-WA1/2020
	SRR11517737-40	Human	human lung biopsies	Infected	9,514,219	USA-WA1/2020
	SRR11412215-18	Human	Lung epithelium NHBE cells	Non-infected	17,003,573	Non-infected (mock)
	SRR11412219-22	Human	Lung epithelium NHBE cells	Non-infected	16,311,121	Non-infected (mock)
	SRR11412223-26	Human	Lung epithelium NHBE cells	Non-infected	24,286,949	Non-infected (mock)
	SRR11412227-30	Human	Lung epithelium NHBE cells	Infected	15,032,096	USA-WA1/2020
	SRR11412231-34	Human	Lung epithelium NHBE cells	Infected	15,108,090	USA-WA1/2020
	SRR11412235-38	Human	Lung epithelium NHBE cells	Infected	44,210,735	USA-WA1/2020
	SRR11412239-42	Human	Lung alveolar A549 cells	Non-infected	27,013,945	Non-infected (mock)
	SRR11412243-46	Human	Lung alveolar A549 cells	Non-infected	14,744,844	Non-infected (mock)
	SRR11412247-50	Human	Lung alveolar A549 cells	Non-infected	11,683,707	Non-infected (mock)
	SRR11412251-54	Human	Lung alveolar A549 cells	Infected	34,141,057	USA-WA1/2020
	SRR11412255-59	Human	Lung alveolar A549 cells	Infected	29,681,064	USA-WA1/2020
	SRR11412260-63	Human	Lung alveolar A549 cells	Infected	20,603,153	USA-WA1/2020
	SRR11517744	Human	Lung-derived Calu-3 cells	Non-infected	9,324,151	Non-infected (mock)
	SRR11517745	Human	Lung-derived Calu-3 cells	Non-infected	17,436,078	Non-infected (mock)
	SRR11517746	Human	Lung-derived Calu-3 cells	Non-infected	37,787,485	Non-infected (mock)
	SRR11517747	Human	Lung-derived Calu-3 cells	Infected	23,623,325	USA-WA1/2020
	SRR11517748	Human	Lung-derived Calu-3 cells	Infected	13,583,713	USA-WA1/2020
	SRR11517749	Human	Lung-derived Calu-3 cells	Infected	28,688,015	USA-WA1/2020
	SRR11517699	Ferret	Trachea	Non-infected	328,105,259	Non-infected (mock)
	SRR11517700	Ferret	Trachea	Non-infected	5,210,254	Non-infected (mock)
	SRR11517701	Ferret	Trachea	Non-infected	4,746,327	Non-infected (mock)
	SRR11517702	Ferret	Trachea	Non-infected	5,163,699	Non-infected (mock)
	SRR11517703	Ferret	Trachea	Infected	9,169,859	USA-WA1/2020
	SRR11517707	Ferret	Trachea	Infected	14,124,547	USA-WA1/2020
	SRR11517711	Ferret	Trachea	Infected	12,933,325	USA-WA1/2020
	SRR11517715	Ferret	Trachea	Infected	14,644,347	USA-WA1/2020
GSE159522	SRR12828440-43	Human	Lung alveolar A549 cells	Non-infected	19,152,790	Non-infected (mock)
	SRR12828444-47	Human	Lung alveolar A549 cells	Non-infected	19,381,530	Non-infected (mock)
	SRR12828448-51	Human	Lung alveolar A549 cells	Non-infected	16,483,541	USA-WA1/2020
	SRR12828428-31	Human	Lung alveolar A549 cells	Infected	17,644,925	USA-WA1/2020
	SRR12828432-35	Human	Lung alveolar A549 cells	Infected	19,504,193	USA-WA1/2020
	SRR12828436-39	Human	Lung alveolar A549 cells	Infected	19,491,861	USA-WA1/2020

**Table 3 cells-10-00319-t003:** MicroRNAs potentially able to bind to the leader sequence of the coronavirus 5′ untranslated region (5′UTR) and their binding energy. Lower binding energy demonstrates higher binding stability between microRNA and leader sequence.

MicroRNA	Organism	Thermodynamic Binding Energy against Leader Sequence (kcal/mol)				
		SARS-COV-2	SARS	MERS	Bat Coronavirus	Bovine Coronavirus
ptc-miR474b	*Populus trichocarpa*	−27.3	−24.5	−21.6	−21.5	−17
ptc-miR474a	*Populus trichocarpa*	−27.3	−22.2	−22.8	−22.2	−18.1
csa-let-7d	*Ciona savignyi*	−25.1	−22.7	−24.6	−24.6	−19
cin-let-7d-5p	*Ciona intestinalis*	−25.1	−22.7	−24.6	−24.6	−19
gga-miR-6608-3p	*Gallus gallus*	−25	−23.6	−30.1	−14.6	−17.1
eca-miR-9080	*Equus caballus*	−23.4	−27.7	−15.9	−15.9	−16.5
csi-miR3953	*Citrus sinensis*	−22.5	−22.4	−27.2	−14.1	−16.8
ame-miR-3741	*Apis mellifera*	−21.9	−20.9	−35.2	−16	−21.7
cel-miR-8207-3p	*Caenorhabditis elegans*	−20.5	−22	−25.6	−12.2	−22.6
ppy-miR-1273a	*Pongo pygmaeus*	−20.1	−21.1	−23.4	−18.7	−19.5
hsa-miR-5004-3p	*Homo sapiens*	−19.4	−25.9	−17.7	−17.7	−14.1
bta-miR-2284ab	*Bos taurus*	−19.3	−21.6	−16.8	−19.9	−13.8
oan-miR-1395-5p	*Ornithorhynchus anatinus*	−19.3	−27.8	−17.5	−15.3	−13.4
mdo-miR-137b-5p	*Monodelphis domestica*	−17.7	−26.8	−19.4	−15.1	−16.8
dme-miR-4949-3p	*Drosophila melanogaster*	−17.7	−17.4	−13.2	−12.6	−24.4
ssc-miR-9833-5p	*Sus scrofa*	−17.1	−16.3	−15.9	−15.9	−15.7
ptc-miR6464	*Populus trichocarpa*	−16.2	−15.4	−13.7	−13.7	−13.1
mtr-miR2629g	*Medicago truncatula*	−15.1	−21.7	−13.1	−13.1	−14.1
mtr-miR2629f	*Medicago truncatula*	−15.1	−21.7	−13.1	−13.1	−14.1
mtr-miR2629e	*Medicago truncatula*	−15.1	−21.7	−13.1	−13.1	−14.1
mtr-miR2629d	*Medicago truncatula*	−15.1	−21.7	−13.1	−13.1	−14.1
mtr-miR2629c	*Medicago truncatula*	−15.1	−21.7	−13.1	−13.1	−14.1
mtr-miR2629b	*Medicago truncatula*	−15.1	−21.7	−13.1	−13.1	−14.1
mtr-miR2629a	*Medicago truncatula*	−15.1	−21.7	−13.1	−13.1	−14.1
bmo-miR-3293	*Bombyx mori*	−13.9	−14.8	−15.7	−12.7	−16
dsi-miR-986-3p	*Drosophila simulans*	−12.4	−16.5	−25.1	−25.1	−19.7
dme-miR-986-3p	*Drosophila melanogaster*	−12.4	−16.5	−25.1	−25.1	−19.7
dsi-miR-986-3p	*Drosophila simulans*	−12.4	−16.5	−25.1	−25.1	−19.7
dme-miR-986-3p	*Drosophila melanogaster*	−12.4	−16.5	−25.1	−25.1	−19.7
mmu-miR-6957-3p	*Mus musculus*	−11.6	−13.5	−12.2	−12.2	−12.1
ppc-miR-83-5p	*Pristionchus pacificus*	−11.4	−14.7	−10.9	−11.4	−17.6
cme-miR1863	*Cucumis melo*	−11.3	−10.5	−12.5	−12.5	−15.2
cel-miR-2211-5p	*Caenorhabditis elegans*	−10.7	−10.6	−10.5	−10.3	−12.2
ath-miR5638a	*Arabidopsis thaliana*	−9.8	−7.8	−10.9	−10.9	−16.6
bdi-miR5065	*Brachypodium distachyon*	−9.7	−11.8	−23.6	−23.6	−16.9
bdi-miR5065	*Brachypodium distachyon*	−9.7	−11.8	−23.6	−23.6	−16.9
oan-miR-1421l-2-3p	*Ornithorhynchus anatinus*	−9.7	−10.8	−12.6	−11.1	−21.2
mghv-miR-M1-2-3p	*Mouse gammaherpesvirus 68*	−9.4	−8.8	−8.3	−5.7	−7.5
dps-miR-2535-3p	*Drosophila pseudoobscura*	−9.3	−10.6	−17.1	−17.1	−19.6
	Average	−16.33	−18.58	−18.34	−16.35	−16.61

**Table 4 cells-10-00319-t004:** hsa-miR-5004-3p genomic variation. High GERP++ conservation score is associated with higher functional impact.

rsId	Chr.	Location	Ref	Alt.	MicroRNA	Gene Region	Translational Impact	GERP++ Score
rs369274154	6	33406128	T	C	MIR5004	5UTR		
rs371304188	6	33406147	C	T	MIR5004	5UTR		
rs375913209	6	33406168	C	T	MIR5004	5UTR		
Not assigned	6	33406194	A	C	MIR5004	5UTR	splice-disrupt	4.77
Not assigned	6	33406194	A	G	MIR5004	5UTR	splice-disrupt	4.77
Not assigned	6	33406194	A	T	MIR5004	5UTR	splice-disrupt	4.77
Not assigned	6	33406195	G	A	MIR5004	5UTR	splice-disrupt	4.77
Not assigned	6	33406195	G	C	MIR5004	5UTR	splice-disrupt	4.77
Not assigned	6	33406195	G	T	MIR5004	5UTR	splice-disrupt	4.77

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
