# Peer review of "A Transcription Regulatory Sequence in the 5′ Untranslated Region of SARS-CoV-2 Is Vital for Virus Replication with an Altered Evolutionary Pattern against Human Inhibitory MicroRNAs"

_cells, 2021, doi:10.3390/cells10020319_

Round 1

Reviewer 1 Report

I have read with interest the study by Manijeh Mohammadi‐Dehcheshmeh et al. The study is uptodate dealing with the very importatn topic. Methods are appropriately used, however some of the results are higly exxagerrated and shal be rephrased not to misslead the reader. Specific commnets can be found bellow.

Major concerns:

  1. Already in abstract authors suggest „low expression of MIR-5004-3p can be con be considered as a COVID19 risk factor“ However, the results the are using to support this finding do not support it as authors only reffere that levels of miR-5004-3p were detected to be lower in samples AFTER COVID infection – this does not place the causality and it can also mean, that levels of miR-5004-3p are lower BECAUSE OF COVID infection, not as it risk factor (i.e. that miR-3004-3p intreacellulary bound to COVID a got degraded). Please, Comment on ths matter and edit manuscript accordingly.
  2. Sentence „Decline in expression of human inhibitory microRNAs with increasing age and diseases can contribute to higher mortality of SARS-CoV-2 in elderly patients“ in abstract is highly speculative, I agree with the authors, but please, introduce the sentence that „We can hypothesize that…“ because this is not a direct finding of your study
  3. Lastly, also statement that „hsa-miR-5004-3p can be considered as a biomarker of severe COVID infection as well as a biomarekr for evaluation of he efficiency of vaccines“is totaly overgerated and such a statemnt cannot be derived from the study showing that hsa-miR-5004-3p is decreased in dengue virus
  4. Discussion: Authors state „In addition to age, human microRNAs downregulate in coronary artery disease, kidney disease, colorectal cancer, osteosarcoma, prostate cancer, obesity/diabetes, myocardial injury, hepatocellular carcinoma, non-small cell lung cancer, gastrointestinal tumors, and colorectal cancer [8].“ However, there are also microRNAs that are upregulated in all mentioned diseases, both in plasma and tissue – please, explain or rephrase.

Minor concerns:

  1. Authors did not follow the manuscript formating not providing page numbers on the left side, thus making it difficult to comment.
  2. Indroction: Authors shall provide 1-2 sentence Introduction about continuous and discontinuous replication patterns as not all the readers will be aware of its meaning
  3. Unify the use of microRNA vs miRNA vs miR abbreviation thorough the manuscript
  4. In methods authors first state that TargetScan was used and then that miRBase was used and both of them were used to retrieve mature sequences (at least) – could you please specify whch information was derived from which databases?
  5. Authors shall correct the Tables order to appear numerically by their appearance in the text
  6. Please, describe how the bufalin is administered – whether is pill, fluid, powder etc.
  7. In concluson authors state „Compared to cellular microRNAs, extracellular microRNAs are highly stable [12,17-19].“ which is not completely true and its a inappropriate mixing of facts. What do the authors want to underline? Please comment.
  8.  

Typos:

There are numerous and numerous typos scattered thorough the manuscript – including innappropriate use of active and passive tense, plural, capital letters… I uploded the pdf with marked changes as it was difficult to locate the typos with lines numbering

Reviewer 2 Report

The study by Mohammadi-Dehcheshmeh et al. attempts to examine the role of the untranslated regions in SARS-CoV-2 genome in the virulence in humans. The authors perform many in silico and biological sample analyses and assert that the transcription regulatory sequence in the 5’ UTR of SARS-CoV-2 plays a role in the virulence. The authors also describe hsa-MIR-5004-3p as an inhibitory microRNA with anti-SARS-CoV2 potential. Based on the finding, the authors propose that decreased levels of functional inhibitory miRNAs contribute to the increased prevalence of COVID-19 is special populations.

The study is timely but has no merit. It lacks the scientific rigor and has overblown statements about its the pioneering findings.  

There are numerous mistakes, spelling and typographical errors making the manuscript difficult to follow. For example, a sentence in the abstract, describing findings:

 “However, its binding stability remarkably decreased in SARS-COV-2 (-19.4 kcal/mol), compared to SARS-COV-2 (-25.9 kcal/mol).

The methods are insufficiently described.

There is no information about origin and ethical consideration (IRB or equivalent) of the biological samples of human and animal origin. These samples were used to examine has-MIR-5004-3p expression. There is no information on the origin of the samples, number of experiments or repeats, normalization methods, internal control used.

Numerous studies showed that the cellular level of a miRNA is not equivalent with its inhibitory potential. Thus, focusing merely on the cellular levels of any miRNA does not validate its role in the COVID-19 infection.  

The section on the repurposing of drugs is based on unsubstantiated and claims rather than scientific evidence. The authors should focus on the validation of their data first before proposing to repurpose drugs.

The discussion and conclusions should be based on the findings of the study.   

Round 2

Reviewer 1 Report

I have read with interest the edited version of the manuscript. The authors answered all my queries and significantly improved the quality of the manuscript. There are only tiny minor comments to be upgraded.

MINOR CONCERNS:

Abstract: Please edit: hsa-MIR-5004-3p was the only human microRNA able to target the leader sequence of SARS and to a lesser extend also SARS-CoV-2.

Figure 1: There is a missing caption for Fig 1 in the current version of the manuscirpt.

Page 11: Please edit: Lack of innate human inhibitory microRNAs able to bind to SARS-COV-2 contributes to the high replication of SARS-CoV-2 in infected human cells.

Conclusion: Please edit: Non-coding regions are crucial for SARS-CoV-2 replication, transcription and domination of host systems biology.

Conclusion: PLease edit: Decline in the expression of human inhibitory microRNAs with increasing age, and in a range of diseases such as diabetes and obesity, can also contribute to higher mortality of SARS-CoV-2 in elderly patients AND IN INDIVIDUALS SUFFERING from THESE DISEASES.

Author Response

Response to Reviewer 1 Comments

Comments and Suggestions for Authors: I have read with interest the edited version of the manuscript. The authors answered all my queries and significantly improved the quality of the manuscript. There are only tiny minor comments to be upgraded.

Response to Comments and Suggestions for Authors: Thank you very much for the positive comments We are very happy that the Reviewer has found that the quality of the manuscript has improved, mentioning all the queries are addressed.

MINOR CONCERNS:

MINOR CONCERN 1: Please edit: hsa-MIR-5004-3p was the only human microRNA able to target the leader sequence of SARS and to a lesser extend also SARS-CoV-2.

Response:

Correction is made according to the mentioned comment. Thank you very much for great comment.

MINOR CONCERN 2: Figure 1: There is a missing caption for Fig 1 in the current version of the manuscirpt.

Response:

Thank you for the point. Correction is made.

MINOR CONCERN 3: Page 11: Please edit: Lack of innate human inhibitory microRNAs able to bind to SARS-COV-2 contributes to the high replication of SARS-CoV-2 in infected human cells.

Response:

Correction is made according to the comment.

MINOR CONCERN 4: Conclusion: Please edit: Non-coding regions are crucial for SARS-CoV-2 replication, transcription and domination of host systems biology.

Response:

Correction is made according to the comment.

MINOR CONCERN 5: PLease edit: Decline in the expression of human inhibitory microRNAs with increasing age, and in a range of diseases such as diabetes and obesity, can also contribute to higher mortality of SARS-CoV-2 in elderly patients AND IN INDIVIDUALS SUFFERING from THESE DISEASES.

Response:

Thank you for the great comment. Correction is made accordingly.